# Effect of Familial Longevity on Frailty and Sarcopenia: A Case–Control Study

**DOI:** 10.3390/ijerph20021534

**Published:** 2023-01-14

**Authors:** Angel Belenguer-Varea, Juan Antonio Avellana-Zaragoza, Marta Inglés, Cristina Cunha-Pérez, David Cuesta-Peredo, Consuelo Borrás, José Viña, Francisco José Tarazona-Santabalbina

**Affiliations:** 1Division of Geriatrics, Hospital Universitario de la Ribera, 46600 Valencia, Spain; 2School of Doctorate, Universidad Católica de Valencia San Vicente Martir, 46001 Valencia, Spain; 3Freshage Research Group, Department of Physiotherapy, Faculty of Physiotherapy, University of Valencia, CIBERFES-ISCIII, INCLIVA, 46010 Valencia, Spain; 4Department of Quality Management, Hospital Universitario de la Ribera, 46600 Valencia, Spain; 5Freshage Research Group, Department of Physiology, Faculty of Medicine, University of Valencia, CIBERFES-ISCIII, INCLIVA, 46010 Valencia, Spain; 6Centro de Investigación Biomédica en Red Fragilidad y Envejecimiento Saludable (CIBERFES), 46010 Valencia, Spain

**Keywords:** aging, function, muscle, interleukin-6, heredity, frailty, sarcopenia, longevity

## Abstract

Familial longevity confers advantages in terms of health, functionality, and longevity. We sought to assess potential differences in frailty and sarcopenia in older adults according to a parental history of extraordinary longevity. A total of 176 community-dwelling subjects aged 65–80 years were recruited in this observational case–control study, pair-matched 1:1 for gender, age, and place of birth and residence: 88 centenarians’ offspring (case group) and 88 non-centenarians’ offspring (control group). The main variables were frailty and sarcopenia based on Fried’s phenotype and the European Working Group on Sarcopenia in Older People (EWGSOP) definitions, respectively. Sociodemographics, comorbidities, clinical and functional variables, the presence of geriatric syndromes, and laboratory parameters were also collected. Related sample tests were applied, and conditional logistic regression was performed. Cases had a higher percentage of robust patients (31.8% vs. 15.9%), lower percentages of frailty (9.1% vs. 21.6%) and pre-frailty (59.1% vs. 62.5%) (*p* = 0.001), and lower levels of IL-6 (*p* = 0.044) than controls. The robust adjusted OR for cases was 3.00 (95% CI = 1.06–8.47, *p* = 0.038). No significant differences in muscle mass were found. Familial longevity was also associated with less obesity, insomnia, pain, and polypharmacy and a higher education level and total and low-density lipoprotein cholesterol. The results suggest an inherited genetic component in the frailty phenotype, while the sarcopenia association with familial longevity remains challenging.

## 1. Introduction

The aging of the world population has become a reality, and the proportion of people aged over 60 years is growing faster than any other age group. In 2018, the population aged 65 years and over exceeded the number of children under 5 years worldwide for the first time in history, and by 2050, the number of people aged 80 years and over will have tripled, from 143 million in 2019 to 426 million (United Nations, Department of Economic and Social Affairs, Population Division (2019). World Population Prospects 2019: Highlights. ST/ESA/SER.A/423). This demographic change is particularly fast in the European Union (EU) as a result of both a longer life expectancy and low fertility rates. The Spanish population aged 65 or over on 1 January 2022 was 9,620,055 people (20.2% of the total population), of whom 16.8% were 85 or older, and 6.4% were 90 years old or more [1]. Spain, with the rapid growth of people over 80 and 90 years old in the last decade, is among the countries with the oldest populations and could have the highest life expectancy in the world by 2040 [2]. Such circumstances could lead to an uncontrollable increase in morbidity and dependency that would compromise the sustainability of the health system. Therefore, increasing research and knowledge on factors related to comorbidity, geriatric syndromes, and disability (to prevent and avoid them), as well as longevity and successful aging (to promote them) should be encouraged.

Frailty and sarcopenia have emerged as two key geriatric syndromes in the aging process and have been associated with adverse events, including falls, fractures, functional impairment, disability, cognitive impairment, hospital admissions, and institutionalization [3,4,5]. These events are likely to limit the quality of life and longevity and increase healthcare costs [6]. Longevity, on the other hand, has been positively associated with a lower frequency of sarcopenia and a delay in the onset of physical frailty and cognitive impairment [7]. Their prevalence may vary depending on the characteristics of different studies, the geographical area, and the gender and age of the participants. In Europe, the prevalence could range between 6% and 27% for frailty [8] and between 1% and 29% for sarcopenia [9]. 

Frailty and sarcopenia, both characterized by the limited capacity of the organism to cope with stressors, are favored by common underlying illnesses and lifestyle and environmental factors [10]. Furthermore, they are dynamic processes, with the possibility of transitions from lower to higher levels and, more difficult, from higher to lower states in the case of frailty [11,12]. The presence of sarcopenia, on the other hand, also appears to modulate transitions in frailty status [13]. Regrettably, both syndromes are frequently detected only when one or both are well established and, often after a seemingly minor event, the individual is already suffering a health crisis with significant functional loss and dependency.

Previous studies have suggested the existence of inheritance patterns in frailty- and sarcopenia-related variables, such as strength, gait speed, and overall physical fitness [14,15,16]. However, the question of how heritability contributes to the onset of these two syndromes remains unanswered.

The delay in functional decline observed in individuals with extraordinary longevity [17] suggests better homeostasis and capacity to adapt and recover from stressors, which might be related to genetic and epigenetic factors, including greater genomic integrity, more preserved methylation, and a characteristic microRNA profile [18,19,20]. The offspring of individuals with favorable genetic characteristics also seem to inherit some advantages from their parents regarding overall health, functionality, and longevity [21,22,23,24,25,26,27,28,29,30]. However, the link between genetic and clinical characteristics in these lineages is poorly understood, thus highlighting the need to investigate heritability in aging-related conditions.

In this study, we used a pair-matched case–control approach to investigate frailty and sarcopenia in community-dwelling persons 65 years and older who are descendants of long-lived individuals and compare them with a population-based control group of age- and gender-matched individuals without a parental history of extraordinary longevity.

## 2. Materials and Methods

### 2.1. Study Design and Participants

This was a cross-sectional matched-control analysis of community-dwelling individuals aged 65 to 80 years who had been born in the area and were usual residents (i.e., >6 months per year) in the Health Department of La Ribera (249,063 people, 71% in small urban areas, Valencian Community, Spain). The study was conducted between 9 March 2015 and 6 February 2017.

Inclusion criteria for cases were as follows: Cases must be 65 to 80 years old, have been born in the study area and lived at least 6 months a year in it, and reside at home (community-dwelling). Finally, they must have at least one parent alive of 97 years or older.

Inclusion criteria for controls were as follows: The control had to be the same age (±5 years) as the case he or she was matched with, have been born in the study area and lived at least 6 months a year in it, and reside at home (community-dwelling). Finally, his or her parents must have died before 90 years old.

Exclusion criteria for both groups were as follows: being diagnosed with a terminal illness, having a life expectancy of fewer than 6 months, or not signing the informed consent form.

The study was conducted according to the principles of the Declaration of Helsinki, and the study protocol was approved by the Research Ethics Committee of the Hospital Universitario de La Ribera de Alzira (Valencia, Spain). Results were reported following the Strengthening the Reporting of Observational Studies in Epidemiology (STROBE) guidelines [31].

### 2.2. Procedures

Three nurses were hired as field researchers. They were specifically trained by the research team in all activities in which they were involved: selecting and recruiting patients; informing patients, relatives, or caregivers and requesting informed consent; organizing and carrying out home visits; handing out questionnaires for medical histories; performing comprehensive geriatric assessments; performing physical and functional tests; measuring bioelectrical impedance; taking blood samples and performing their initial processing; and transferring the samples to the laboratory. After the training process, it was verified that the field researchers had a high degree of agreement.

Study candidates were selected by the field researchers following a three-stage procedure. First, the entire population database of the Health Department was screened for individuals aged 97 years or more. Candidates were contacted (either directly or via their relatives/caregivers) and informed about the study. After obtaining oral informed consent, a home visit was scheduled in which candidates were asked about living offspring aged 65 to 80 years and permission to contact them. Second, offspring candidates were contacted, informed about the study, and invited to enroll. Candidates who provided written informed consent were included in the study as cases unless they met any exclusion criteria. Together with the inclusion of cases, a population-based matched control group was established by pairing 1:1 for gender, age (±5 years), and place of birth and residence. Controls were screened among individuals in the population registry of the Valencian Community. The same eligibility criteria, except that their parents must have deceased before 90 years old, and the same procedure for obtaining informed consent were used. During home visits, members of both groups underwent the same physical evaluation, questionnaires, functional and cognitive tests, bioelectrical impedance measurement, and blood draws.

The initial protocol can be freely consulted on the internet at https://www.educacion.gob.es/teseo/mostrarRef.do?ref=1943613# (accessed on 30 December 2022).

### 2.3. Primary Outcomes

The two co-primary outcomes were frailty and sarcopenia. Frailty was assessed based on Linda Fried’s phenotypic criteria [32]: unintentional weight loss (>4.5 kg in the last year), exhaustion (the subject reported that any activity was too strenuous or he/she was unable to continue carrying it out at least 3 days in the previous week), low physical activity (weekly time walked was 2 h/week in women and 2.5 h/week in men [33]), weakness (grip strength, measured with a JAMAR© analog dynamometer, below the cut-off point, stratified by body mass index (BMI) and gender), and slowness (15-foot (4.572 m) gait speed above the cut-off point, stratified by gender and height, or inability to complete the test). Individuals were considered frail if they met 3 or more criteria; pre-frail if they met 1 or 2; and robust if none of the criteria were met.

The presence of sarcopenia was assessed according to the 2018 European Working Group on Sarcopenia in Older People (EWGSOP) definition [34]. Muscle mass was assessed by electrical bioimpedance analysis (BIA) using a four-sensor measurement device (OMRON BF500© manufactured by OMRON. ’s-Hertogenbosch, The Nederlands). For low muscle quantity, a cut-off point of 7 kg/m^2^ in men and 6 kg/m^2^ in women was adopted. Muscle strength was determined by measuring the grip strength with JAMAR©, and cut-off points of 27 kg in men and 16 kg in women were used. Muscle performance was measured by gait speed in 4.572 m; the cut-off point was 0.8 m/s in both genders.

### 2.4. Secondary Variables

Sociodemographic and clinical variables were also collected to better characterize the sample, such as educational level, smoking, alcohol intake, medications, physical exercise, nutritional risk (DETERMINE scale), the performance of instrumental (Lawton and Brody index) or basic activities of daily living (Barthel index), limitations in physical activity or disability (modified Rankin scale), mobility ability (Holden’s Functional Ambulation Classification (FAC)), the presence of comorbidities (age-adjusted Charlson comorbidity index), cognitive status (Spanish adaptation of the Mini-Mental State Examination, MEC-Lobo), depression and anxiety (Goldberg depression and anxiety scale), falls in the last 3–6 and 12 months, pain (visual analog scale), quality of life (Spitzer Quality of Life Index), social assessment (OARS scale), resource consumption, blood pressure, weight in kilograms, and height in centimeters.

Laboratory parameters included fasting glucose, lipid profile, renal and hepatic profile, C-reactive protein (CRP), interleukin-6 (IL-6), testosterone, sex-hormone-binding globulin (SHBG), thyroid-stimulating hormone (TSH), 25-hydroxycholecalciferol (25-OH D), iron profile, vitamin B12, folic acid, and complete blood count.

### 2.5. Statistical Analysis

Variables were assessed for normality using the Kolmogorov–Smirnov test. Quantitative variables were described using the mean and standard deviation (SD) and the median and interquartile range (IQR, defined as the difference between 75th and 25th percentiles) for normally and non-normally distributed variables, respectively. Categorical variables were described as frequency and percentage (%). Tests for paired data were used in order to investigate differences between groups. McNemar’s test was applied for categorical variables with 2 × 2 contingency tables. In the case of quantitative variables with a normal distribution, Student’s *t*-test for related samples was applied to compare means between the two groups. The Wilcoxon signed-rank test for matched samples was applied when a quantitative variable did not follow a normal distribution, and also in the case of ordinal categorical variables with more than two categories [35,36]. For the study of the whole sample, encompassing both groups together (i.e., some tables in Appendix A), tests for independent samples were applied. Specifically, Chi-squared for categorical variables, Pearson’s correlation for quantitative variables, and the Kruskal–Wallis test for quantitative variables that did not meet the normality assumption. The adjusted odds ratio (OR) for robustness was assessed using a conditional logistic regression (CLR) that included all variables potentially confounding the effect. We considered a variable to be a candidate for CLR if it was associated with familial longevity and with robustness in controls with a *p*-value less than 0.2 and caused a change in the OR equal to or greater than 10% [37]. A sensitivity analysis was performed for the serum levels of IL-6, CRP, ferritin, leukocytes, and lymphocytes. We took into consideration that some of the individuals included in the case group could be siblings. Thus, significant differences between siblings and nonsiblings in the case group were assessed for age, gender, BMI, comorbidity, frailty, muscle mass, gait speed, handgrip strength, sarcopenia, inflammatory parameters, and lipids. Data processing and statistical analysis were performed with SPSS Statistics for Macintosh v21 (Armonk, NY, USA: IBM Corp) and Stata 16.0 for PC.

## 3. Results

### 3.1. Participant Disposition and Characteristics

Sixty-three long-lived people were interviewed, resulting in 96 offspring being contacted for eligibility. After considering the inclusion and exclusion criteria, 88 cases and 88 controls were analyzed (Figure 1). Sixty percent of the recruited samples were female, and the median age was 70 years (IQR = 7). The most prevalent comorbidity was hypertension (58%), followed by dyslipidemia (43.2%) and fractures in any location (36.4%) (Appendix A: Comorbidities). The most frequent geriatric syndromes were polypharmacy (38.1%), followed by insomnia (34.9%), pain (34.7%), depression (21.6%), and falls within the last year (21.1%) (Appendix A: Geriatric Syndromes).

Table 1 shows the main demographic, clinical, and functional characteristics. Compared with cases, individuals in the control group had a significantly higher weight (with a higher obesity prevalence), had a higher frequency of insomnia and pain, and were prescribed more drugs, with a higher percentage of polypharmacy. The two groups also differed in their education levels, with a lower percentage of university degrees and a higher percentage of the non-completion of primary education in controls than in cases. No significant differences were observed between the groups in smoking, alcohol intake, exercise/physical activity, nutritional risk, scores on the different functionality scales, falls, depression, anxiety, cognitive status, quality of life, social resources, and health service utilization. The age-adjusted Charlson index did not reveal significant differences in the overall comorbidity burden; however, in individuals over 70 years, the score was significantly lower in the case group (median = 3; IQR = 1) than in controls (median = 4; IQR = 1) (*p* = 0.022) (Appendix A: Charlson index).

Regarding analytical variable measurements (Table 2), cases presented significantly higher levels of total and LDL cholesterol. The percentage of individuals with an LDL/ApoB ratio lower than 1.3, which is associated with the presence of small and dense LDL particles, was significantly higher in the control group. Cases showed higher levels of SHBG and lower levels of IL-6 than controls. No significant differences were observed in the other variables analyzed. 

Thirty-eight individuals from the case group were siblings (43.2%), and no statistically significant differences were found between sibling and nonsibling cases for age, gender, BMI, comorbidity, frailty, muscle mass, gait speed, handgrip strength, sarcopenia, inflammatory parameters, and lipids.

### 3.2. Frailty and Sarcopenia

Overall, 27 (15.3%) individuals in the study sample were frail, and 107 (60.8%) were pre-frail. The prevalence of frailty and pre-frailty was significantly higher in women: 17% and 65.1%, respectively, among women vs. 12.9% and 54.3% among men (*p* = 0.023). Reduced physical activity and weakness were the most prevalent frailty criteria (55.4% and 50.3%, respectively). According to EWGSOP criteria, 4.6% of the overall sample had sarcopenia, and 1.7% had severe sarcopenia. The overall prevalence of sarcopenia was higher in women (9.4%) than in men (1.4%) (*p* = 0.009).

The percentage of robust patients was significantly higher among cases (31.8%) than controls (15.9%) (*p* = 0.001). Likewise, lower percentages of frail and pre-frail individuals were observed in the cases (9.1% and 59.1%, respectively) than in the control group (21.6% and 62.5%, respectively) (Figure 2A). Of all frailty criteria, weakness, slowness, and exhaustion were significantly less prevalent in cases than in controls (Table 3). The crude OR of cases for robustness was 3.33 (95% IC = 1.38–8.06, *p* = 0.009). The attributable fraction in the case group was 70% (95% IC = 23–90%) and 22% in the population (95% = IC 7–35%). After the confounding assessment, the number of medications and blood levels of 25-hydroxycholecalciferol were selected. Both variables were associated with robustness in controls and with familial longevity, with a *p*-value < 0.2, and caused a change in the OR equal to or greater than 10%. After conditional logistic regression had been performed, the robust adjusted OR for cases was 3.00 (95% IC = 1.06–8.47, *p* = 0.038).

When using the 2018 EWGSOP criteria, we found no significant differences in sarcopenia between groups (Figure 2B). Cases showed lower percentages of weakness and slowness; however, no significant differences were observed regarding muscle mass (Table 3).

## 4. Discussion

In this pair-matched case–control observational study, a familial history of parental longevity was associated with greater robustness and lower frailty in their 65–80-year-old offspring. However, the longevity history did not significantly contribute to the presence of sarcopenia due to the lack of differences in muscle mass. Besides being more robust, descendants of long-lived individuals had a higher education level, a lower prevalence of obesity, lower levels of IL-6, higher total and LDL cholesterol (with a lower percentage of subjects with an LDL/ApoB ratio of less than 1.3), and a lower prevalence of geriatric syndromes such as pain, insomnia, and polypharmacy.

The prevalences of frailty, pre-frailty, and robustness in the total sample, as well as the percentage of women, were within the ranges reported in the existing literature [8]. Previous research on the history of familial longevity and frailty has considered the latter either with a definition based on the accumulation of deficits or with a phenotypic definition. In our opinion, the frailty index could be more appropriate for assessing more complex geriatric patients, while phenotypic categorization may be more adequate for younger, less complex older people or those at risk of frailty [38,39], such as those in our sample. Kim et al. and Arosio et al., with the first approach, found that family longevity was significantly associated with a lower rate of frailty [40,41]. Nevertheless, research using a phenotypic definition has not always presented similar results. In the Long-Life Family Study, with the Scale of Aging Vigor in Epidemiology (SAVE) [42] based on Fried’s criteria, moderate heritability and less frailty in offspring were found. On the other hand, The LonGenity study concluded that, though the offspring of parents with exceptional longevity had better measures of physical function, there were no differences in frailty using Fried’s criteria [43]. The results obtained in our study could be explained by a different design, with younger participants from a different geographic location and, presumably, with greater genetic heterogeneity than the aforementioned study. Furthermore, the cases we studied had significantly lower levels of IL-6, and this may suggest a lower degree of age-related inflammation, consistent with their lower degree of frailty [44,45].

Three other geriatric syndromes that were less prevalent in centenarians’ offspring were insomnia, pain, and polypharmacy. Sleep is one of the processes affected by aging, with lower sleep efficiency and REM sleep [46], and older adults with more ADL and IADL limitations have a higher risk of experiencing a decline in sleep quality [47]. A certain degree of heritability for sleep–wake patterns and insomnia has been suggested [48,49], but the mechanisms involved are still under investigation, and its association with longevity is controversial. In fact, the empirical support for an increase in mortality risk with insomnia is inconsistent, although it might be higher with the use of hypnotic medication [50], and previous research did not show significant differences in sleep patterns between centenarians’ offspring and their controls [51]. In contrast, our familial longevity case group had significantly less insomnia than their matched controls. The relationship between sleep disturbances and frailty status seems more evident [52]; however, the implications of these associations are not well established. Insomnia may be a marker for different conditions, such as poor health, comorbidities, disabilities, social impairment, or, interestingly, inflammatory cytokines. Both conditions, frailty and insomnia, might share some underlying mechanisms. Thus, we understand that the differences observed for insomnia support the consistency of the results found for frailty and familial longevity in the present study.

Although no significant differences were found between groups in anxiety, depression, cognition, falls, fractures, or osteoarthritis (Appendix A), our control group reported significantly more pain than offspring enriched for human longevity. Pain is a complex experience that involves sensory-discriminative, affective-motivational, and cognitive-evaluative dimensions. When investigating the relationships of parental longevity with the regional brain structure, some differences in zones involved in transferring and processing sensory and nociceptive information, which might support our results, have been found, but more research is needed on this issue [53]. 

Regarding pharmacological therapy, a lower use of medications has been observed previously in descendants of centenarians [28], which could be related to their lower comorbidity incidence, better physical and cognitive function, and health perception. Nevertheless, differences in the Charlson index were observed in our study, being statistically significant only in people over 70 years of age. Thus, this difference in comorbidity after the age of 70, in addition to the higher prevalence of pain and insomnia in the control group, and their probable need for treatment might have influenced differences in polypharmacy.

Some metabolomic and lipidomic studies have been published, trying to identify a specific profile for aging, longevity, and frailty [54,55,56,57]. However, this goal was beyond our scope. The metabolic profiles of our patients were evaluated with traditional analytical variables, such as the lipid profile, fasting glucose and others (Table 2), weight, and BMI. Except for total and LDL cholesterol and BMI, no significant differences were observed between the case group and the controls.

The relation between cholesterol, frailty, and longevity is complex. Higher levels of total and LDL cholesterol have been linked to cardiovascular disease (CVD), and it is also known that centenarians have lower levels than younger people. Nevertheless, in some studies [58,59], higher concentrations have been also associated with centenarians’ offspring, theoretically enriched for longevity, and less frail elderly. We also found this apparently less favorable lipid profile (i.e., higher levels of total and LDL cholesterol) in our case group with familial longevity. However, in this regard, it is worth mentioning that small, dense low-density lipoproteins seem to be a better marker for cardiovascular disease outcomes [60]. We did not have access to any of the different laboratory techniques used to separate LDL fractions into subfractions. Notwithstanding the above, we found that the control group, despite having lower cholesterol levels, more frequently showed an LDL/ApoB ratio of less than 1.3, which suggests the greater presence of small and dense LDL particles and, therefore, a higher cardiovascular risk for controls [60,61]. This result is in line with a previous study [58], which suggested that the offspring of long-lived individuals, despite presenting higher levels of total and LDL cholesterol, could have a better profile in LDL particles, which might explain the lower incidence of cardiovascular disease reported in previous research [62]. Hypercholesterolemia is frequently associated with obesity and overweight, and a positive correlation of total and LDL cholesterol levels with age and BMI is known, especially up to age 65 [63]. In our sample, total and LDL cholesterol showed a slight (less than 0.3), but statistically significant, negative correlation with BMI (Appendix A). However, because of the small magnitude of the coefficient and the loss of significance (except for LDL) when stratified by case group, we cannot draw any relevant conclusions from this result.

Differences in carbohydrate metabolism, associated with lower insulin resistance and a lower percentage of diabetes, with family longevity have been extensively reported in previous studies [22,23,25,26,27]. Along this line, we found that the case group showed a nonsignificant statistical tendency (*p* = 0.08) to present less diabetes and altered basal glucose, which could reflect an underpowered sample to detect differences. Notwithstanding the above, previous studies also found no significant differences in this regard [28,64]. Thus, we should also consider that different characteristics of the studied populations, in terms of environmental factors, habits, lifestyle, and diabetes prevalence, might have contributed to these results.

Previous research has suggested some degree of heritability for both muscle mass and function [15,65], lower morbidity and mortality, and slower functional loss in centenarians’ offspring [24,64], raising the possibility that they may also have a lower prevalence of sarcopenia. Still, in the present study, no significant differences were detected, probably due to the low prevalence found using the EWGSOP criteria [66]. Likewise, it is worth mentioning that this result was due to the lack of differences in the muscle mass index between groups, while muscular performance, as assessed by handgrip strength and gait speed, was clearly better in the offspring of long-lived individuals. This lack of correlation might suggest that the relationship between mass and function could vary depending on factors related to the population studied [67].

Chronic inflammation and obesity have been linked to frailty in previous research. In the present study, familial longevity, in addition to higher robustness and better muscular performance, was also significantly associated with a lower prevalence of obesity and lower levels of inflammation, as measured by Il-6 levels. This raises the question of whether these differences might confound the results. It is worth mentioning in this regard that neither BMI nor IL-6 levels were significantly associated with robustness or frailty in the global sample (Appendix A). Therefore, statistically, they do not meet the confounding criteria. Previous research suggests that both frailty and obesity might be influenced by genes and underlying mechanisms related to inflammation and energy metabolism [57,68,69,70]. Thus, we understand the lower inflammation, BMI, and frailty prevalence found in our case group are more likely a question of internal consistency than a source of a potential confounding effect.

In contrast to previous studies [28,43], no significant differences were observed in the cognitive and functional profiles of the two groups. Similarly, no differences were observed between the groups in nutritional status, toxic habits, exercise, and physical activity. Existing studies have found that centenarians’ offspring have a lower incidence of comorbidities, later disease onset, lower polypharmacy, and a lower mortality risk than their peers without a history of family longevity [21,25,28]. In the present study, the offspring of long-lived individuals showed a lower polypharmacy prevalence and a lower Charlson index score, but we only observed this last result when the analysis was conducted in the 70+ age group (Appendix A), suggesting that the difference in comorbidity is likely to be more evident at older ages. This might be a consequence of a slower pace of aging, which would be in line with previous research [26,71].

The biological mechanisms that could explain the advantages for frailty and muscle performance observed in centenarians’ offspring compared with their controls are still under investigation. Centenarians have specific characteristics that lead to the deceleration of the aging rate throughout their lives [72]. Along this line, in previous research, they showed the differential expression of mRNA and miRNA related to processes, such as cellular damage protection and the modulation of the immune response, associated with healthy aging and frailty [20,73]. Additionally, it has been observed that some special traits of centenarians might be inherited by their offspring, distinguishing them from non-centenarians’ offspring [57]. The former could possess better-preserved metabolic patterns in order to face the increase in energy demand associated with the maintenance of homeostasis, health status, and functionality. Furthermore, in a subsample of subjects from the present study, we found that offspring overexpressed genes related to bone growth activation, muscle development, skeletal development, and cell differentiation [74], which could partly explain the differences found in frailty and muscle performance.

Some limitations in our study should be considered. Its cross-sectional nature increases its susceptibility to bias, as it is not conducive to the establishment of causal relationships. The study population was geographically restricted to our Health Department, and thus, the number of candidates for the study was limited. Assuming the robustness prevalence found and the matched case–control groups with a paired data design, our sample size would have been enough for 85% power. Its matched design allowed us to control for important variables, such as age, gender, birthplace, and residence, as well as increase efficiency in obtaining identical sample sizes [75]. 

We must consider the possibility that some variables may have influenced our results, confounding the association between familial longevity and frailty. In relation to this, it is worth mentioning that a study variable could have a confounding effect on the results if it satisfies the three following properties: there must be statistically significant differences between cases and controls regarding the variable studied, it must have a statistically significant association with frailty, and it must not be an effect of familial longevity (intermediate variable) [76]. The last criterion is usually the most complex to address, so the process normally begins with the verification of the first two, and if they are met, it proceeds to the verification of the third criterion.

Some lifestyle habits can influence the onset and progression of frailty [77]. Regarding the presence of habits such as smoking, alcohol intake, or regular physical exercise in our study, no significant differences were observed between the case group and the controls (Table 1), and no association was found between these variables and frailty (Appendix A). The results for physical activity were somewhat different. Although no significant differences were found between the case group and the control group (Table 1), participants with moderate to vigorous physical activity were significantly more robust and less frail than those with mild activity in the whole sample (Appendix A). This contrast between regular physical exercise and physical activity might be explained by better accuracy when we classify the level of physical activity according to the usual tables ([78]) with respect to the generic question of whether or not a person exercises regularly. After these considerations, we consider it unlikely that differences in these habits could have influenced our results in a decisive way.

The relation between body weight and frailty is still under investigation, with the probability that the underweight BMI category has a higher risk [79]. As previously mentioned in this discussion, although BMI was significantly higher in the control group, there was not a significant association between BMI and frailty in our overall sample (Appendix A). On the other hand, despite the low prevalence of sarcopenia observed, we also explored the presence of sarcopenic obesity and its possible influence on our results. Only 11 participants met the EWGSOP criteria for sarcopenia; none of them were obese, and in both groups, the majority were in the normal-weight category (Appendix A). Therefore, despite the important relation between BMI, sarcopenic obesity, and frailty demonstrated in previous research, it seems unlikely to us that differences in weight, or sarcopenic obesity, could have a relevant confounding effect on our results.

Additionally, no differences in other potentially confounding variables, such as nutrition and physical activity, were found in the bivariate analysis. Furthermore, two variables that met the criteria for confounding were identified, and the conditional logistic regression carried out supports our findings. Therefore, it seems probable that the extent of the genetic enrichment effect in the case group, besides the study design, might explain the ability to find significant differences in the frailty phenotype. We should also consider that some degree of nonresponse bias could be possible due to the unknown characteristics of the controls who refused to participate. This bias occurs when key characteristics of the study make a difference between respondents and nonrespondents. In previous research, respondents had higher education levels and reported better health and satisfaction levels than nonrespondents [80,81]. These aspects, if they occurred in our study, could have reduced our power to detect differences rather than the opposite, thus highlighting our results. Additionally, it should be noted that the use of BIA to assess muscle mass might have influenced the results for sarcopenia. BIA has some advantages, such as portability and ease of use, but on the other hand, it offers less validity than magnetic resonance imaging (MRI), computerized tomography (CT), or dual-energy X-ray absorptiometry (DXA). In addition, using this device requires precise instructions to minimize errors. Regarding this, it is worth mentioning that the field researchers were trained both in this technique and in how to conduct the tests and implement questionnaires in order to try to control potential sources of bias.

Finally, we have to consider that the population studied was clearly delimited and belonged to an area of the Spanish Mediterranean coast, which might limit the external validity of the results. Notwithstanding the above, and although long-lived families may also have healthier living habits or better socioeconomic conditions, the benefits associated with family longevity appear to be independent of such factors and may be largely attributable to a genetic influence [82]. In this way, we observed a 22% attributable fraction in the population, which is similar to the Robustness Index Ratio observed by Serena Dato et al. in the Longitudinal Study of Aging Danish Twins [6].

## 5. Conclusions

In summary, our findings indicate that the offspring of long-lived individuals have significantly lower odds of developing frailty within the age range of 65 to 80 years compared with an age- and gender-matched control group. This finding suggests an inherited component of the frailty phenotype, consistent with the lower levels of markers of underlying inflammatory processes and the better muscular performance. However, results regarding the prevalence of sarcopenia in this group of offspring are controversial, and the enhanced muscular performance conflicts with the lack of significant differences in muscular mass.

## Figures and Tables

**Figure 1 ijerph-20-01534-f001:**
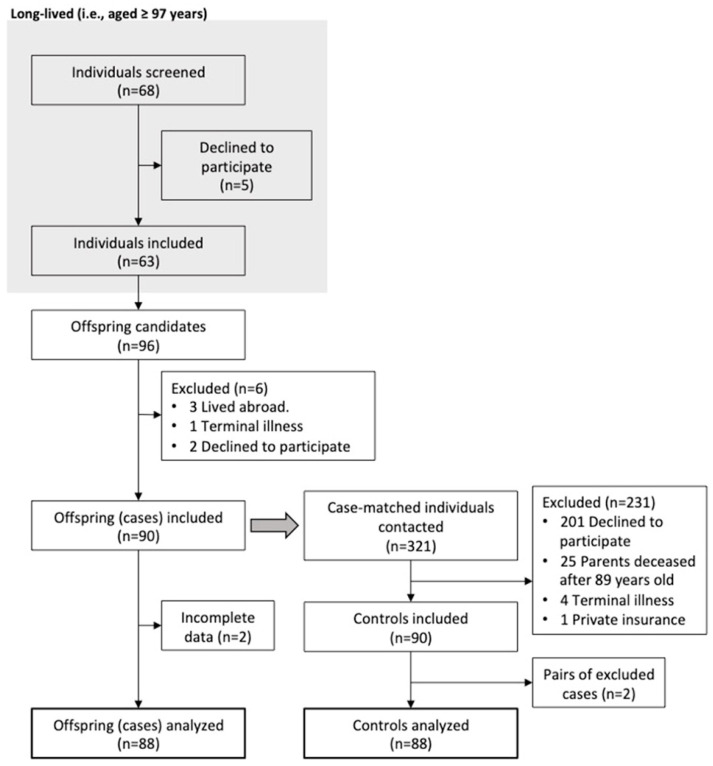
Flow chart of subjects’ selection: shaded rectangle = selection process of long-lived; arrow pointing downward = progress flow; arrow pointing to the right = exit flow; gray arrow = start of control selection.

**Figure 2 ijerph-20-01534-f002:**
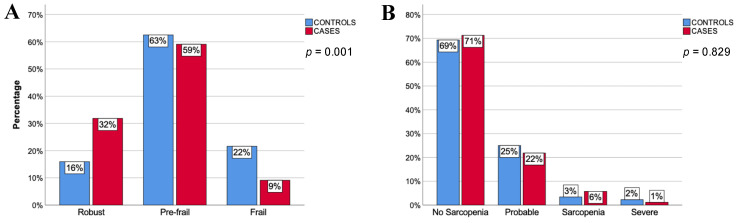
Results for frailty (panel **A**) and sarcopenia (panel **B**): Y-axis = percentage of subjects within group; X-axis = categories; blue columns = controls; red columns = cases; *p* = statistical significance.

**Table 1 ijerph-20-01534-t001:** Demographic, health, and functional characteristics.

	Overall n = 176	Controls n = 88	Cases n = 88	*p*-Value ^a^
Age, years, median (IQR)	70 (7)	69 (7)	70 (6)	0.69
Female, n (%)	106 (60)	53 (60)	53 (60)	1
Weight, kg, mean ± SD	74.1 ± 14.4	76.9 ± 13.6	71.2 ± 14.6	0.008 *
BMI, kg/m^2^, median (IQR)	28.1 (5.6)	29.2 (6.1)	27.6 (4.5)	0.007 *
Underweight, n (%)	5 (2.9)	1 (1.1)	4 (4.6)	0.004 *
Normal weight, n (%)	30 (17.1)	12 (13.6)	18 (20.7)	
Overweight, n (%)	79 (45.1)	35 (39.8)	44 (50.6)	
Obese, n (%)	61 (34.9)	40 (45.5)	21 (24.1)	
Current smoker, n (%)	23 (13.1)	12 (13.6)	11 (12.5)	0.424
Alcohol use, n (%)	46 (26.6)	26 (29.9)	20 (23.9)	0.472
Mild physical activity ^b^, n (%)	96 (56.1)	52 (59.1)	44 (53%)	0.417
Regular exercise ^b^, n (%)	123 (69.9)	64 (72.7)	59 (67)	0.372
Nutritional risk ^c^, n (%)	44 (25)	23 (26.1)	21 (23.8)	0.819
Sleeping hours				
Hours per day, median (IQR)	7 (2)	7 (2)	7.5 (1)	0.640
Insomnia, n (%)	59 (34.9)	37 (44.6)	22 (25.6)	0.030 *
Charlson, median (IQR)	3 (2)	3 (2)	3(2)	0.350
Lawton, median (IQR)	8 (0)	8 (0)	8 (0)	0.719
Barthel, median (IQR)	100 (0)	100 (0)	100 (0)	0.305
Rankin ≤1, n (%)	170 (97.7)	84 (96.6)	86 (98.8)	0.613
FAC ≥4, n (%)	133 (95.7)	69 (94.5)	64 (97)	0.461
QL Spitzer, median (IQR)	10 (1)	10 (0)	10 (1)	0.839
Falls ≥1/12 months, n (%)	34 (21.1)	18 (22.2)	16 (19.8)	0.148
Pain ^d^, n (%)	61 (34.7)	38 (43.2)	23 (26.1)	0.029 *
Goldberg scale				
Anxiety, n (%)	33 (18.8)	21 (23.9)	12 (13.6)	0.137
Depression, n (%)	38 (21.6)	18 (20.5)	20 (22.7)	0.850
MMSE-Lobo, median (IQR)	32 (4)	33 (3)	32 (5)	0.592
Polypharmacy ^e^, n (%)	67 (38.1)	43 (48.9)	24 (27.3)	0.003 *
Nº of drugs, median (IQR)	3 (3)	3 (5)	2 (3)	0.001 *
OARS good or excellent, n (%)	172 (98.3)	85 (96.6)	88 (100)	0.554
Lives alone, n (%)	8 (4.5)	5 (5.7)	3 (3.4)	0.727
Education, n (%)				0.034 *
No primary	24 (13.8)	16 (18.2)	8 (9.3)	
Primary	108 (62.1)	52 (59.1)	56 (65.1)	
Secondary	28 (16.1)	17 (19.3)	11 (12.8)	
University	14 (8)	3 (3.4)	11 (12.8)	
Healthcare utilization, medical visits/year, median (IQR)				
Primary care	2 (3)	2 (3)	2 (2)	0.644
Specialized care	1 (2)	1 (2)	0 (2)	0.518

Notes: Mean and standard deviation (SD) for normal quantitative variables. Median and interquartile range (IQR) for non-normal quantitative variables; n = patients; % = percentage; BMI = body mass index; FAC = functional ambulation classification; QL = quality of life; MMSE = Mini-Mental State Examination; OARS = Older Americans Resources and Services; ^a^ = McNemar test for categorical variables, Student’s *t*-test for related samples for quantitative variables with normal distribution, and Wilcoxon signed-rank test for matched samples for non-normal quantitative variables and ordinal categorical variables of more than 2 categories; ^b^ = 150 min per week or more; ^c^ = 3 points or more on Determine scale; ^d^ = 2 days per week or more; ^e^ = regular use of at least five medications; * *p* < 0.05.

**Table 2 ijerph-20-01534-t002:** Laboratory results.

	Overall n = 176	Controls n = 88	Cases n = 88	*p*-Value ^a^
Glucose, mg/dL, median (IQR)	95 (25)	96 (27)	93 (21)	0.113
<100, n (%)	108 (61.4)	49 (55.7)	59 (67)	0.084
100–125, n (%)	40 (22.7)	22 (25)	18 (20.5)	
≥126, n (%)	28 (15.9)	17 (19.3)	11 (12.5)	
Albumin, g/dL, median (IQR)	4.4 (0.3)	4.4 (0.3)	4.3 (0.3)	0.301
Transferrin, mg/dL, mean ± SD	277.3 ± 40.8	281.3 ± 46.2	273.3 ± 34.4	0.506
Ferritin, ng/dL, median (IQR)	89 (101)	85 (110.5)	89 (96)	0.274
Total cholesterol, mg/dL, mean ± SD	199.1 ± 39.5	191.7 ± 37.3	206.4 ± 40.4	0.015 *
LDL, mg/dL, mean ± SD	115.7 ± 35.3	110.1 ± 32.1	121.3 ± 37.5	0.043 *
HDL, mg/dL, median (IQR)	56.5 (20)	55.5 (18)	58 (22)	0.213
VLDL, mg/dL, median (IQR)	23 (14)	23 (13)	22.5 (14)	0.370
Triglycerides, mg/dL, median (IQR)	113 (70)	115 (68)	112 (72)	0.360
ApoB, mg/dl, mean ± SD	101.5 ± 20.6	98.9 ± 21.3	104 ± 19.7	0.141
LDL/ApoB Ratio < 1.3, n (%)	135 (77.1)	74 (84.1)	61 (70.1)	0.038 *
Vitamin B12, pg/mL, median (IQR)	372 (188)	376 (188)	371 (191)	0.549
Folic acid, ng/mL, median (IQR)	10.5 (6.7)	11.3 (7.1)	9.4 (6.7)	0.360
25-OHD, ng/mL, mean ± SD	19.6 ± 6.5	20.3 ± 6.5	18.8 ± 6.5	0.193
TSH, mcU/mL, median (IQR)	1.59 (1.20)	1.48 (1.12)	1.67 (1.07)	0.562
Total testosterone, ng/mL, median (IQR)	0.58 (3.47)	0.60 (3.13)	0.57 (3.67)	0.328
Free testosterone, ng/mL, median (IQR)	0.08 (0.05)	0.009 (0.05)	0.008 (0.05)	0.334
SHBG, nmol/L, median (IQR)	50.3 (26.2)	47.6 (23.1)	56.1 (33.1)	0.004 *
Hemoglobin, g/dL, mean ± SD	14.4 ± 1.2	14.3 ± 1.2	14.4 ± 1.3	0.502
Leukocytes, ×10^9^/L, mean ± SD	6.72 ± 1.79	6.84 ± 1.89	6.60 ± 1.70	0.475
Lymphocytes, ×10^9^/L, median (IQR)	1.80 (0.75)	1.90 (0.80)	1.80 (0.78)	0.733
CRP, mg/L, median (IQR)	1.49 (2.34)	1.92 (2.24)	1.18 (2.14)	0.155
IL-6, pg/mL, median (IQR)	1.20 (1.26)	1.45 (1.38)	1.03 (0.96)	0.044 *

Notes: Mean and standard deviation (SD) for normal quantitative variables. Median and interquartile range (IQR) for non-normal quantitative variables; n = patients; % = percentage; LDL = low-density lipoprotein cholesterol; HDL = high-density lipoprotein cholesterol; VLDL = very-low-density lipoprotein cholesterol; ApoB = apolipoprotein B; 25-OHD = 25-hydroxycholecalciferol; TSH = thyroid-stimulating hormone; SHBG = sex-hormone-binding globulin; CRP = C-reactive protein; IL-6; interleukin-6; ^a^ = McNemar test for categorical variables, Student’s *t*-test for related samples for quantitative variables with normal distribution, and Wilcoxon signed-rank test for matched samples for non-normal quantitative variables and ordinal categorical variables of more than 2 categories; * *p* < 0.05.

**Table 3 ijerph-20-01534-t003:** Frailty and sarcopenia components.

	Controls n = 88	Cases n = 88	*p*-value ^a^ n = 88
Frailty, n (%)			
Weight Loss	5 (5.7)	6 (6.8)	1
Exhaustion	15 (17)	4 (4.5)	0.013 *
Reduced activity	54 (61.4)	43 (49.4)	0.082
Weakness	50 (56.8)	38 (43.7)	0.045 *
Slowness	22 (25)	9 (10.3)	0.004 *
Sarcopenia, n (%)			
Low muscle mass	8 (9.4)	10 (11.8)	0.815
Weakness	27 (30.7)	25 (28.7)	0.038 *
Slowness	28 (32.6)	14 (16.3)	0.018 *

Notes: ^a^ = McNemar test; * *p* < 0.05.

## Data Availability

Data analyzed in the current study are available from the corresponding author upon reasonable request.

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
