# Peer review of "Effect of Familial Longevity on Frailty and Sarcopenia: A Case–Control Study"

_ijerph, 2023, doi:10.3390/ijerph20021534_

Round 1

Reviewer 1 Report

The paper is well written and data presented appear solid. However, it lacks originality. furthermore, it would be appreciable a more extensive discussion about metabolic implications of decreased body weight and BMI, as well as increased total cholesterol and LDL levels, in the case group. Are these findings correlated? Is there a specific metabolic and anthropometric pattern in older patients compared to the general population? what about sarcopenic obesity?

Author Response

Dear reviewers and editor in chief of The International Journal of Environmental Research and Public Health:

The authors want to thank you for giving us the opportunity to submit a revised draft of this manuscript to the International Journal of Environmental Research and Public Health. We appreciate the effort that you and the reviewers have dedicated to providing us your valuable feedback on our manuscript. The authors will answer one item at a time following the order established by the reviewers in the evaluation. All the revisions are marked in the article, using de Track Changes function with MS Word

Answer to reviewer #1

We have expanded the discussion about results in BMI, total cholesterol and LDL levels, and their correlation, and we have also introduced a comment on sarcopenic obesity as Reviewer #1 suggested. Also, the authors have included some paragraphs  (in the main text)  and some tables (in the supplemental material) to comment and clarify the possible influence of habits, lifestyle, physical activity and BMI in our results, as Reviewers  #1 and #3 suggested.

Reviewer 2 Report

In the manuscript, titled "Effect of familial longevity in frailty and sarcopenia: A case-control 2 study," the authors used a pairwise matched case-control approach to assess frailty and sarcopenia in individuals aged 65 years and older, offspring of long-lived individuals, to examine and compare them to a population with no parental history of exceptional longevity. The manuscript is generally well and concisely written and the study design is optimal. However, this reviewer suggests that the visuals could have been better. Important parameters could have been presented as box plots/error bar plots (indicating statistical significance). For example, in the supplemental figure displaying the Charlson index (Figure S1), differences between matched case-control pair histograms (A and B) should be combined with two color choices, with common X and Y axes indicating the differences. The criterion for the chosen parametric and non-parametric statistical analysis (p-values) must be explained in detail.  Remove the Authors  degree qualifications("MD, Ph.D.") below the title of the manuscript.

Author Response

Dear reviewers and editor in chief of The International Journal of Environmental Research and Public Health:

The authors want to thank you for giving us the opportunity to submit a revised draft of this manuscript to the International Journal of Environmental Research and Public Health. We appreciate the effort that you and the reviewers have dedicated to providing us your valuable feedback on our manuscript. The authors will answer one item at a time following the order established by the reviewers in the evaluation. All the revisions are marked in the article, using de Track Changes function with MS Word

Answer to reviewer #2

Title

Authors degree qualifications below the title of the manuscript have been removed, as Reviewer #2 suggested.

Methods

Instead of mentioning the different statistics tests that we had listed before performing the statistical analysis, in the new version of the manuscript we only explain the tests that were finally applied and why. We have also changed the footnotes in the respective tables. We believe that these changes improve understanding of the criteria used to choose the statistical tests, in the way that Reviewer #2 has suggested.

Supplemental material

We have included some new tables in the supplemental material, to support the commentaries added to the discussion.

We have modified visuals for Charlson index by adding a new visual  (Figure S2) and modifying  the Figure S1 (Charlson index) to improve its understanding,  following the suggestions of Reviewer #2.

Reviewer 3 Report

Manuscript shows the relation between sarcopenia and frailty in elderly and their descendants. I have some comments for you.

Abstract: EWGSOP: should be explained the first time you mention it.

Introduction: concise and well-explained, but too short, maybe you should add more data about the importance these pathologies have, the prevalence of people older than 90 years in the country… to support and justify your study.

Methodology: a procedure section should be added to explain how, who, when and where data was collected.

Discussion: although habits, lifestyle, physical activity is not the key point of your study, some information about these aspects of the sample could influence your results, considering the body mass above all. Also, if the older people life in their home or if they are institutionalized, are data that must be taken into account. Have you considered these issues?

References do not follow mdpi rules, please correct them.

Author Response

Dear reviewers and editor in chief of The International Journal of Environmental Research and Public Health:

The authors want to thank you for giving us the opportunity to submit a revised draft of this manuscript to the International Journal of Environmental Research and Public Health. We appreciate the effort that you and the reviewers have dedicated to providing us your valuable feedback on our manuscript. The authors will answer one item at a time following the order established by the reviewers in the evaluation. All the revisions are marked in the article, using de Track Changes function with MS Word

Answer to reviewer #3

Abstract

The EWGSOP definition has been added in the abstract after the first time is mentioned, as Reviewer #3 suggested.

Introduction

We have expanded the introduction section by adding more information about demographics, sarcopenia and frailty, as Reviewer #3 suggested to support and justify our study.

Methods

The authors have added a procedures subsection for better understanding how, who, when and where data was collected, and we have emphasized the fact that all de participants in the study were community-dwelling and not institutionalized, as Reviewer #3 has suggested. We have also added a link to the full report that includes the initial protocol of the study, wich is available in the web page of the Ministry of Universities of the Spanish Governemnt.

Discussion

We have expanded the discussion about results in BMI, total cholesterol and LDL levels, and their correlation, and we have also introduced a comment on sarcopenic obesity as Reviewer #1 suggested. Also, the authors have included some paragraphs  (in the main text)  and some tables (in the supplemental material) to comment and clarify the possible influence of habits, lifestyle, physical activity and BMI in our results, as Reviewers  #1 and #3 suggested.

References

References

We have modified the references to properly follow the mdpi rules, as Reviewer #3 suggested, and some references have been added to support some of the changes we have made to the text.

Round 2

Reviewer 3 Report

Dear authors, 

Much better, thank you for considering my suggestions.